# A New Approach for Evaluation True Stress–Strain Curve from Tensile Specimens for DC04 Steel with Vision Measurement in the Post-Necking Phases

**DOI:** 10.3390/ma16020558

**Published:** 2023-01-06

**Authors:** Sławomir Świłło, Robert Cacko

**Affiliations:** Faculty of Mechanical and Industrial Engineering, Institute of Manufacturing Technologies, Warsaw University of Technology, 02-524 Warszawa, Poland

**Keywords:** vision extensometer, uniaxial tensile testing, necking geometry, image processing, stress–strain curve correction

## Abstract

The paper presents an experimental evaluation of deformation of flat samples during uniaxial tensile testing, including uniform deformation and post-necking phases. The authors recommend a specially designed vision extensometer and simplified image processing method for analytical correction of triaxial test results for extended stress–strain curve estimation. A modified correction model is proposed, based on the application of Gaussian functions, to determine the neck geometry of the tested sample. The vision extensometer can monitor a specimen’s elongation using two fibre-optic gauges inserted into the material. Measurements taken from the vision extensometer are compared with readings from analogue gauges within the range of uniform deformation. The analytical correction model’s ability to correctly assess the extended true stress–strain curve in the post-necking phase was investigated. Image processing forms the basis of an efficient method for identifying the contour of the specimen’s neck. Digital image correlation (DIC) was used to verify the proposed solutions and assess the results obtained for the uniform and post-neck deformation phases. The change in thickness of the sample was experimentally measured throughout the tensile test with a digital gauge sensor and compared with the results of the digital image correlation.

## 1. Introduction

Uniaxial tensile testing is one of the most basic mechanical test methods for characterising elastic and plastic deformation behaviour. Testing conducted under the conditions outlined by the standards PN-91/H-04310 or PN-EN 10002-1+AC1 determines the mechanical properties of materials widely used in many technological areas. Tensile testing may be conducted to determine the modulus of elasticity (Young’s modulus), limits of plasticity and elasticity, yield strength, ultimate strength, upper and lower yield points, stress at fracture, elongation at rupture, relative elongation, and total elongation [1].

The rapid growth of manufacturing technology in various industries has led to a growing need for more accurate and practical methods to test materials. New technical solutions are continually being devised for uniaxial tensile testing, mainly because of developments in physics and optoelectronics. A quick and accurate analysis of uniaxial tensile test results is expected to allow characterisation of several mechanical properties of a material related to elastic and plastic deformation and a detailed evaluation of plastic hardening. Knowledge of these material properties is essential to ensure that materials meet the stress and strain requirements for use in modern industrial systems. The quality of the testing data must be high to allow comparison with computer simulation results by the finite element method (FEM) or other methods of analysis to generate complex mechanical models of cracks, fatigue, stress distributions, and deformations [2]. Any of several strength and deformation parameters may be crucial to the performance of sheet metal products obtained by forming processes, such as doors, hoods, and automobile body components [3].

Traditional deformation measurement during uniaxial tensile testing is conducted using special measuring devices, such as capacity strain extensometers or contact and optical extensometers, up to the point of necking. Once that state of triaxial stresses is reached, other measuring techniques are necessary to account for the change in the cross section during necking. Therefore, a combination of two measurements, standard (elongation of the gauge length) and extended (cross-sectional area of the gauge length), is usually considered to obtain the extended formula for the true stress–strain curve.

The digital image correlation (DIC) method is known to be a valid assessing tool in mechanics experimental procedure. It is one of the most widespread methodologies for studying strain in diverse materials [4,5,6]. It is applied in solving various challenges in materials science [7,8], solid mechanics [9], civil engineering [10,11,12,13], etc. It has to be mentioned, that other techniques applied recently for mechanical testing analysis were used successfully, like Acoustic Emission Technique (AET) [14] or micrography and morphology [15,16].

The authors demonstrate an enhanced method for evaluation of the true stress–strain curve using a vision extender and image processing (VEIP) technique. The proposed solution allows for monitoring of the elongation of the gauge length during the uniform deformation phase, based on changes detected by fibre-optic lighting (FOL) gauges inserted into a flat sample. This solution is crucial for supplementing the initial course of measurements with accurate representation of the sample’s elongation. In the post-necking phase, image processing is used to continue the initial displacement measurements based on contour image processing identification (CIPI) analysis. Proper setting of the vision extensometer is essential to correctly calibrate and capture images obtained during tensile testing, including the necessary unit conversion (i.e., pixels to mm). This solution results in converted images at short time intervals (incremental steps of deformation) which are compiled to produce the extended true stress–strain curve. However, the post-necking phase of the true stress–strain curve is evaluated using an analytical correction model (ACM) based on the approach proposed in this paper. The authors assumed that the neck length was the same in both cross sections perpendicular to the direction of the tensile load. The geometry was examined by means of a CIPI analysis of the sheet surface to define the neck length. A modified correction formula is proposed for the a/R ratio, where a is half of the specimen thickness and R is the radius of the neck, in the Gaussian contour fitting (GCF) process and the specimen’s thickness calculation.

The authors propose a procedure for verifying the results obtained for each phase of the uniaxial tensile test. First, in the uniform stage, the deformation process is monitored in real time using the proposed fibre-optic lighting system with additional digital and analogue sensor gauges. Second, the analytical correction formula is applied after necking, based on a new concept for specimen contour fitting. Third, the strain components (longitudinal and latitudinal) are verified using digital image correlation. Finally, the thickness change is demonstrated using a specially designed digital sensor and combined with the DIC results.

### 1.1. Extensometer Measurement Techniques

In recent years, a wide range of methods have been developed for displacement measurement, using clipped extensometers, lasers, and vision instruments to assess strain deformation [17]. These methods involve complicated procedures and advanced technical solutions and are rarely used for industrial purposes. Furthermore, experimental evaluation can only be performed for uniform deformations (under uniaxial tension) measured to calculate the true stress–strain curve.

Various extensometers were presented in a comprehensive overview of popular methods for the evaluation of the properties of tested materials [18]. This review covered the range of applications of these measuring devices and details of their use during tensile testing. According to the review, the devices may be classified as contact or non-contact devices. According to the review’s authors, when selecting a suitable extensometer, one should consider the specimen’s material, geometry, and testing requirements, as well as the method of data transmission and the testing location. Clip-on extensometers and sensor arms are in direct contact with samples and thus cause minor damage during use and reduce the accuracy of readings. In recent years, non-contact devices have become increasingly efficient due to the use of optoelectronics. This group of devices includes those that measure deformation by recording images of marks applied to the sample’s surface. Appropriate illumination of the surface during deformation is critical to the success of this technique. Inadequate illumination could lead to incorrect readings caused by incorrect assessment of changes in the positions of applied marks during testing. Another non-contact method involves the use of the latest form of laser extensometers, which take advantage of structural changes that occur on the sample’s surface. Laser light directed at the surface is reflected in various directions, causing a unique discolouration pattern corresponding to the magnitude of the material’s deformation. This method does not require permanent marks on the sample.

Non-contact methods offer a wide range of solutions that are continually being improved using the newest forms of laser innovation in this rapidly developing technological field. These include Moiré interferometry, an optical technique for measuring area with high accuracy which is used to track the progress of uniaxial tension tests [19]. These methods require highly advanced optoelectronic systems, such as laser diodes, collimated systems, and mirrors in imaging optics. The use of laser interferometer-based non-contact extensometer measurement is often confined to laboratory testing purposes because of the degree of complexity that would be involved in adapting them to industrial conditions [19]. These devices are usually integrated with testing machines and operate under the machines’ standard working conditions. This method allows for the evaluation of two displacement components in the plane in which the sample is tested to determine the area of stress, strain, and rotation, and to predict trends in deformation patterns characteristic of various phases of tensile testing. It allows for measurements during static, monotonic, and cyclical loading and a complete analysis of any sequence of images [20].

Another vision technique group comprises solutions requiring advanced numerical computations and image vision registration during testing. The basis of this measurement method is a model of regular patterns imposed on the tested surface, as proposed by Sirkis [21]. In addition, a three-dimensional reconstruction is conducted using images of the tested sample recorded by two charged–coupled device (CCD) cameras at any moment during testing. The tested sample requires a stochastic grid used with spray paint to generate the location and shape digitally. This method is based on the numerical optical comparison of images from two cameras at any time. This system divides the image into many small sections and computes the local stress tensor to within a margin of error of 1 mm [22]. This solution allows the collection of more data concerning the sample gauge length (for non-uniform deformations) during uniaxial tensile testing. As a result, numerical simulations can be supplemented with information regarding the magnitude and distribution of stresses in areas of potential cracks [23,24].

### 1.2. Theory of the Tension Test

#### 1.2.1. Uniform Deformation

A stress–strain curve is commonly used to describe fundamental mechanical properties related to the strength and deformation behaviour of materials. Uniaxial tensile testing involves measurement of the load and elongation of a flat (e.g., sheet metal) or round (cylindrical) sample. The unit deformation of the specimen, also known as relative strain or engineering strain, is calculated as the ratio of the change in gauge length over the initial gauge length, as shown below:(1)$$\varepsilon_{eng}=\frac{\Delta L}{L_{0}},$$
where ∆*L* is the change in gauge length after testing and *L*_0_ is the initial length of the gauge and.

Similarly, engineering stress is calculated as the ratio of the applied load to the initial cross-sectional area of the sample, as shown below:(2)$$\sigma_{eng}=\frac{P}{A_{0}},$$
where *P* is the load applied to the sample and *A*_0_ is the initial cross-sectional area of the specimen.

However, to calculate the true strain in the sample, one needs to integrate the strain, Equation (1), according to the incremental strain formula, assuming that it would increase proportionally:(3a)$$\varepsilon_{t}=\int_{L_{o}}^{L}\frac{\Delta L}{L_{0}}=ln\frac{L}{L_{0}}=\ln\left(1+\varepsilon_{eng}\right),$$
where *L* is a successively measured value of the specimen gauge and *ε_t_* is the logarithmic strain.

The engineering stress can be converted to the true stress by considering the instantaneous cross-section of the specimen:(3b)$$\sigma_{t}=\frac{P}{A},$$
where *A* is the instantaneous cross-sectional area of the specimen.

The effective stress and strain can be obtained using the Huber–Mises formula in the Cartesian coordinate system, as shown below:(4a)$$\varepsilon_{e}=\sqrt{\frac{2}{3}\;\;}\sqrt{\varepsilon_{1}^{2}+\varepsilon_{2}^{2}+\varepsilon_{3}^{2}},$$
(4b)$$\sigma_{e}=\frac{\sqrt{2}}{2}\sqrt{{\left(\sigma_{1}-\sigma_{2}\right)}^{2}+{\left(\sigma_{2}-\sigma_{3}\right)}^{2}+{\left(\sigma_{3}-\sigma_{1}\right)}^{2}},$$
where *ε*_1_*, ε*_2_*,* and *ε*_3_ are the principal strain components in the longitudinal, latitudinal, and thickness directions, respectively; *σ*_1_ is the principal stress component in the longitudinal direction; and 1, 2, and 3 are the longitudinal, latitudinal, and thickness directions, respectively for the tensile test (flat sample). The sum of the principal strain components is equal to zero (plastic incompressibility), and a direct relationship between the components can be written as follows:(5)$$\varepsilon_{1}+\varepsilon_{2}+\varepsilon_{3}=0\;\;\;\;\;\;⟹\;\;\;\;\varepsilon_{2}=\varepsilon_{3}=\frac{1}{2}\varepsilon_{1},$$

Substituting these equations into Equation (4a) yields the following equations:(6)$$\varepsilon_{e}=\varepsilon_{1}\;\;\;and\;\;\sigma_{e}=\sigma_{1},$$

Therefore, the uniaxial tensile test within the uniform deformation range is carried out under the conditions of Equation (6), and it can be concluded that the stress–strain curve *σ*_1_ (*ε*_1_) is a generalised formula for the tension test results. A stress–strain diagram can be computed from the recorded load–displacement data as a simple graphical representation of the uniaxial tension test results. These data can then be fitted using one of the analytical formulae [25,26,27] for strain hardening. The analytical yield model developed describes the relation of true stress to true strain using two parameters, the strength coefficient (*K*) and the strain hardening exponent (*n*), as follows:(7)$$\sigma_{p}=K\cdot \varepsilon^{n},$$

The analytical yield model, Equation (7), is frequently used as a simple physical interpretation of plastic behaviour. Direct input of flow stress can also be converted into this symbolic form to model plastic hardening in a computer simulation (e.g., FEM). However, the physical restriction of the tensile test geometry (material necking) limits this approach to low strain levels. Therefore, it is necessary to predict and analytically describe large plastic deformation in a post-necking analysis of flow stress behaviour.

The following four phases may be distinguished in a uniaxial tensile test. The specimen first undergoes simple uniform deformation, during which the uniaxial stress calculation model can be applied for the engineering stress–strain curve evaluation (Equations (1)–(5)). In this phase of the tensile test, the change in width is proportional to the difference in thickness (Figure 1a). When the maximum load is exceeded, the strain becomes localised, geometric instability starts to occur, and the specimen’s contour changes into the shape of a neck. This phase (Figure 1b) is called diffuse necking [28] and leads to triaxial stresses, making it difficult to determine the true stress–strain curve. As the tension test progresses, only the localised neck continues to stretch, while the remainder of the specimen retains its previous geometry. This state is known as sheet metal instability (Figure 1c). Finally, material instability is achieved, and the cracking process leads to fracture (Figure 1d). The phases that specify the material deformation location can be described by mathematical notation. In the initial stage of deformation, the sample begins to experience diffuse necking when the following condition is satisfied:(8a)$$d\left(\sigma_{p}\cdot A\right)=0,$$
where *A* is the cross-sectional area of the sample and *σ_p_* as the plastic stress.

Similarly, for the first derivative of the Hollemon [27] yield analytical model, Equation (7), satisfaction of the condition under which the onset of necking occurs leads to the corresponding strain *ε* for the maximum load:(8b)$$\frac{d\sigma_{p}}{d\varepsilon }=\sigma \;\;\;⟹\;\;\frac{\varepsilon^{n-1}}{n}=\varepsilon^{n}\;\;\;⟹\;\;\varepsilon =n,$$

The neck is localised as the tension test continues, and the thickness of the material sheet decreases rapidly along the specified line (Figure 1c). Localised necking depends on the thickness of the sample and runs along the line formed by an approximately 55° angle with the axis of the sample. Since the length of the line is constant, it can be assumed that the applied force is proportional to the product of the thickness and the plastic stress:(9a)$$d\left(\sigma_{p}\cdot g\right)=0,$$
where *g* is the thickness of the specimen.

Therefore, for the true stress–strain curve, the first derivative satisfies the condition under which the onset of localised necking occurs, according to the following formula:(9b)$$\frac{d\sigma_{p}}{d\varepsilon }=\frac{\sigma_{p}}{2}\;\;\;\;\;⟹\;\;\varepsilon =2\cdot n,$$

Finally, loss of stability begins once the plastic stresses reach their extreme values, (Figure 1d), which can be stated as follows:(10)$$\frac{d\sigma }{d\varepsilon }=0.$$

The four phases of the tensile test described above (uniform deformation, onset of necking, neck localisation, and fracture) are illustrated by the load–deformation curve shown in Figure 2. The triaxial stresses must first be examined to define the true stress–strain curve beyond uniform deformation.

A geometric evaluation, to calculate the smallest cross section of the specimen, follows, which can be described using the following notation:(11)$$\sigma_{e}=C\cdot \sigma_{ave}=C\;\cdot \frac{F}{A},$$
where *C* is a correction factor and *σ_ave_* is the average stress, defined as the ratio of the applied tensile force *F* to the minimum cross-sectional area *A* of the sample.

Issues that can arise in determining the extended stress–strain curve include the localisation of the neck during the sheet metal instability phase and the possibility of non-symmetrical necking. These can lead to difficulties in correctly determining the specimen’s stresses and cross-sectional area. Therefore, the extended stress–strain curve can only be successfully specified up to the moment when localised necking occurs (Figure 1c). The authors reviewed analytical methods for characterising triaxial stress deformation to identify an appropriate way to determine the strain correction factor. The VEIP technique was developed for use in CIPI analysis and to evaluate the cross-sectional area of the specimen at any moment during testing. Finally, the calculation of the stress–strain curve was verified using the DIC technique, in which the material deformation was examined using a stochastic grid applied to the specimen’s surface.

#### 1.2.2. Extended Stress–Strain Curve

Since the mid-20th century, many researchers have investigated the correction procedure for the strain–stress curve. Mathematical investigations of tensile necking in cylindrical and flat bar specimens under triaxial stresses have been conducted by many researchers, such as Onat [29], Bridgman [30], Siebel [31], and Davidenkov [32], who were the first to formulate an ACM for stress and strain states in the post-necking phase, and, more recently, Kaplan [33] and Zheng [34]. The solutions proposed in these studies were based on the assumption of plane strain deformation in the cross section of a tested plate or sheet specimen. Plane strain deformation occurs when the maximum force and concentration of strain causing the neck formation are reached and triaxial stress occurs. The material flow in the neck area is constrained in the direction perpendicular to the edge of the sample. It is assumed that the material displacement is parallel to one plane, i.e., independent of the 3-axis (Figure 3a).

This assumption is valid for plates, where the width of the specimen is not restricted, but its adoption for flat sheets has been highly debated [35] since the subsequent necking process disrupts the plane strain state (Figure 1b–d). Based on experimental data, Bridgman assumed that the minimum stress is uniformly distributed at the neck. The Onat equation [29] used throughout this study incorporates corrections for the computation of true stress beyond the constant strain by applying the linearisation of yield conditions (substitution of the Tresca yield), as well as a material plane:(12)$$\sigma_{1}-\sigma_{2}=2\cdot {\bar{\tau }}_{o},\;\mathrm{and}$$
(13)$$\varepsilon_{1}+\varepsilon_{2}=0.$$

For simplicity, the proposed solution assumes that the stress distribution of the specimen is reduced by two shallow symmetrical grooves. The application of linearisation (Equations (12) and (13)) yields the following:(14)$$\varepsilon_{z}=\varepsilon_{xy}=0\;\;\;\;\;\;and\;\;\;\;\;\sigma_{z}=\sigma_{xy}=0,$$

Equation (14) is justified by forming imperfections in the form of long shallow grooves during the initial necking phase. In the geometry adopted for the necking process, during which the sample is weakened by two symmetric trenches (Figure 4), the following simple function is proposed to describe the shape of the neck:(15)$$y=\pm a+c\cdot x^{2},$$
where 2*a* is the minimum width of the specimen and *c* is a parameter related the geometric properties of the neck. The axial load across any section is normalised to the thickness and is found to be the following:(16)$$F=2\int_{0}^{a}\sigma_{x}dy=4a\cdot k\left(1+a\cdot c\right),$$

Finally, for a flat specimen, converting the average stress within the neck area and substituting the radius and thickness, the following equation for the correction factor can be obtained:(17)$$C_{1}=2\cdot a{\left(1+a\cdot c\right)}^{-1}={\left(1+\frac{a}{2\cdot R}\right)}^{-1},$$

However, a sufficient neck depth is the main criterion for determining stress and strain field changes in tensile testing of thin samples. The specific limitation requirement for neck geometry is unknown, and parabolical functions described by Equation (15) result in the assessment that the length of grooves should be at least 4a [29].

Similarly, as in the formula proposed by Onat [29], supplementary correction formulae can be found in the literature. The main difference in the procedure adopted for stress correction arises from the proposed geometric descriptions. In Bridgman’s research [30], the curve is described by two parameters: the radius of the neck contour and half of the thickness, a. Bridgman similarly employed linearisation of the yield condition, as well as material plane plastic flow, and developed the following formula:(18)$$C_{2}={\{{\left(1+\frac{2\cdot R}{a}\right)}^{\frac{1}{2}}\cdot \ln\left(1+\frac{a}{R}+\left(\frac{2\cdot a}{R}\right)+{\left(1+\frac{2\cdot R}{a}\right)}^{\frac{1}{2}}\right)-1\}}^{-1},$$
where *t* is the thickness of the neck and *R* is the radius of the neck.

The formation of imperfections justifies the application of linearisation and simplification in the form of long shallow grooves during the initial phase (diffuse necking). The practical aspects of measurement motivated additional research to develop new formulae and new gauging techniques. Bridgman’s ACM for stress correction was successfully incorporated into the round bar tension test [36]. However, modifications continue to be proposed for flat specimens, and new formulae are still being investigated. The main challenge in these investigations is the configuration of the neck geometry in the final stage of the tension test (neck localization). The final neck contour is less deformed, smooth, asymmetrical, and challenging to identify. In calculating the extended stress curve for the Bridgman solution, many investigators have faced the problem of accurately characterising the necking geometry. The main reason for this is that some methods are complicated and not applicable in practice [37]. The correction factor in the Bridgman analytical solution is based on parameters that need to be evaluated for the neck profile of the gauge length (i.e., the curvature and radius) for each moment in time.

Zhang [38] suggested an inverse method to evaluate the minimum cross-sectional area as a modification of Bridgman’s formula, based on an extensive three-dimensional numerical simulation analysis of the behaviour of the neck and the relationship between cross-sectional reduction and thickness reduction. The inverse method involves determining the stress–strain curve from the load–thickness relation in the diffuse necking zone. However, it is challenging to measure the neck contour accurately.

Brunet [39] proposed the following true stress–strain evaluation approach based on the length of the neck:(19)$$C_{3}={\left[1+\ln\left(1+\frac{t\left(t_{o}-t\right)}{l^{2}}\right)\right]}^{-1}C_{2},$$
where *t_o_* is the initial thickness of the flat specimen and *l* is the characteristic neck length.

In addition to investigating new or modified analytical formulae for the correction of stress–strain curves, researchers have investigated new gauge techniques, along with numerical methods, especially digital image processing. These methods are based on quick and accurate numerical analysis of the test data. The most practical methods involve computer-aided techniques, with fully automated procedures to capture and analyse hundreds of images. Precise examination of the neck geometry requires efficient algorithms for image processing. Two-dimensional digital image correlation (DIC) for the stochastic and Moiré fringe, which increases the accuracy of the results, has resolved many of the challenges identified [40]. For example, the Siebel correction model uses DIC and image processing [23]. This approach is described in more detail below because of its similarity to the approach taken in the present study. The calculation is based on the necking model for a cylindrical sample, but it was applied and experimentally tested with a flat sample. The proposed Siebel model is similar to the Onat model (Equation (17)), which, after the modification, takes the following form:(20)$$C_{4}=\frac{4\cdot R}{4\cdot R+w}={\left(1+\frac{w}{4\cdot R}\right)}^{-1},$$
where *R* is the radius and *w* is the smallest width of the neck.

A flat specimen with a stochastic pattern was analysed using the DIC technique in that investigation. Additional image processing was used to accurately measure the necking’s shape (e.g., radius and width) at each moment of the deformation process. The images were captured and analysed, and the correction for post-neck deformation was applied according to Equation (20).

Finally, the authors presented all the previously mentioned solutions for the correction factors and combined them in Figure 5. It can be seen that the most significant parameter of the analytical methods is the ratio *a*/*R*, which needs to be computed using complex measurements. Le Roy et al. [41] proposed the following empirical expression to represent the geometric parameters of the neck:(21)$$\frac{a}{R}=1.1\left(\varepsilon -\varepsilon_{p}\right),$$
where *ε* is the logarithmic longitudinal strain and ε*_p_* is the maximum uniform deformation. The above equation was applied to flat samples, and the results were satisfactory [38].

#### 1.2.3. Evaluation of the Shape of the Sample

In this paper, the authors propose an alternative solution for evaluating the shape of the sample. It was assumed that the neck length was the same in both cross sections perpendicular to the direction of tensile load (Figure 3c). Therefore, the neck length must be measured accurately based on the specimen sheet. A modified correction formula was proposed which involves the ratio *a*/*R* and the measured thickness of the sample. The authors recommend the application of a Gaussian function to fit the data (GCF process). An advanced image processing technique (described in the next section) was developed to estimate the neck contour (CIPI analysis). Since the Gaussian function depends on a simple parameter, the neck shape could be computed easily using the following formula:(22)$$y=c\cdot \exp\left(\frac{-{\left(x-\mu \right)}^{2}}{2\cdot b^{2}}\right)+h,$$
where *µ* is the horizontal shift of the neck within the coordinate image system, *b* is the neck characteristic (99.7% of the values lie within ± 3*b*), *c* is the depth of the neck, and *h* is the vertical shift (Figure 6).

In the experiment conducted, the authors assumed that the neck length was the same in both cross sections perpendicular to the direction of the tensile load (Figure 3c). Thus, for a flat specimen, the geometry of the neck was measured perpendicularly to the sheet metal instead of in the strain plane (Figure 3b), where the radius is difficult to determine. Next, to simplify the measurement procedure for neck identification, a standardised parameter *b* characterising the length of the neck, based on a Gaussian function, was used in place of the radius of the neck *R*.

Based on the geometric relations and curvature calculations, an alternate representation can be introduced in the form of *a*/*R*, which appears in most studies according to the following notation:(23)$$R=\frac{\frac{\partial^{2}y\left(x\right)}{\partial x^{2}}}{{\left(1+{\frac{\partial y\left(x\right)}{\partial x}}^{2}\right)}^{\frac{3}{2}}}=\frac{b^{2}}{c}=\frac{2\cdot b^{2}}{t_{u}-t},$$

This leads to a new representation of the relation:(24)$$\frac{a}{R}=\frac{t}{2\cdot R}=\frac{t\left(t_{u}-t\right)}{4\cdot b^{2}},$$
where *t_u_* is the thickness of the sample due to the maximum strain at uniform deformation and *t* is the continuous change of the diffuse neck (Figure 6). The solution in Equation (24) is very similar to Brunet’s Equation (19), with the accuracy improved by the proposed magnification factor.

## 2. Materials and Methods

### 2.1. Vision Extensometer Techniques

A measuring technique using machine vision for length measurements in the uniaxial tensile testing evaluation was employed (Figure 7a). For this purpose, a vision system equipped with a camera and positioning system (for the calibration of the camera) was designed (Figure 7b). The essence of the designed system was the simple option of measuring the elongation of the initial length of the tensile sample, known as the measuring base (Figure 7c).

The proposed method generates a real-time evaluation of a flat sample extension, which is a significant limitation for methods such as DIC, which require post-computational processes. Additionally, low cost and a simple solution were obtained for the method by using fibre optics powered from a LED light source. In combination with a high accuracy measurement of gauge length, using image processing that can be utilised to detect the two-dimensional position of a light spot simultaneously [42], the proposed techniques can be used as an alternative to traditional strain gauge methods, especially in the area of industrial technolgy.

### 2.2. Experimental Procedure

A series of tests were carried out on DC04 steel, commonly used in the automotive industry. The chemical composition of the material is summarised in Table 1. Sample sheet plates with a nominal thickness of 1 mm were manufactured using punching technology and die-blanking tools. A schematic illustration of the sample geometry with dimensional characteristics is presented in Figure 8a. The direction of the tensile load was parallel to the rolling direction.

All samples were measured according to the descriptions in Figure 8b. All samples were equipped with FOL gauges, and stochastic grids were sprayed on their surfaces. All tests were conducted using a standard tensile test machine at cross-head speed. For the designed uniaxial test, the force was recorded by the load cell, located in the tension machine frame, where the specimen was gripped. Three different techniques for measuring the deformations of the tested samples were used. The first consisted of holding a digital gauge sensor at the crosshead (Figure 9a). The second was the proposed VEIP technique, using FOL gauges (Figure 9b). The third involved the use of an analogue gauge to verify the accuracy of the proposed VEIP technique (Figure 9c). A complete set of load and displacement results was taken for each sample. The load response was normalised to the longitudinal specimen deformation to directly compare the three measurement techniques.

The measurements were recorded and analysed in the Matlab/Simulink environment with image processing and data acquisition toolboxes. The first measurement technique (Figure 9a) demonstrated typical errors and issues related to contact extensometers, such as the rigidity of the machine grip on the specimen. For the proposed VEIP technique, 1-mm-diameter fibre-optic cables were mounted into holes drilled through the sample. The small size of the wires meant that the loss of sample material was negligible and did not affect the sample behaviour during the test.

Additional testing of the deformation around the holes was conducted using the DIC technique to demonstrate the lack of influence on the surrounding area. FOL gauges were illuminated through a power supply system of two separate LEDs (Figure 9b). The proposed VEIP technique was used with a vision calibration procedure described in the following section. An analogue gauge was used for the final type of measurement (Figure 9c), with a small measurement base and manual data registration. The precision of the analogue gauge permitted verification of the accuracy of the proposed solution.

The experimental measurements allowed accurate values of stress and strain during uniform deformation to be determined according to Equation (3a). True stress was defined as the ratio of applied load to the immediate cross-sectional area of the deformed sample (Equation (3b)). To obtain an extended stress–strain curve (beyond uniform deformation), it was necessary to apply a multiplication factor C, obtained using one of the analytical solutions described previously (Equations (12)–(15)).

### 2.3. Image Processing—Thresholding

An image process known as thresholding is a crucial process for classifying pixels. The primary aim of this process is to classify the pixels of the analysed image into two groups: those referring to the object and those referring to the background. The threshold level can be determined globally (considering the whole picture) or locally (selecting an appropriate window size). In terms of identifying points signifying the base length of the sample and its geometric outline, it was decided that the global approach would be sufficiently accurate, quick, and straightforward. This technique determines the threshold based solely on the greyscale of the pixels represented by the histogram of their statistical distribution and is independent of the greyscale of surrounding pixels.

Therefore, given image *A*, a set of picture pixels, one could determine the appropriate threshold *t* by minimising the image histogram for the probability distribution obtained. With *Z^+^* defined as the set of all positive integers and (*x*, *y*) as the coordinates of pixels in a digitalised image, *A*, *Z*^+^ = {0, 1, … *I* − 1} is the set of grey levels, where *l* is the total number of quantisation levels. Any pixel’s brightness function (i.e., grey levels) can be defined as *h*(*x*, *y*). For an image depicted in this way, the threshold level *t ϵ Z*^+^, where the set *S* = {*a*_1_,*a*_2_} is two points defining grey levels and belonging to the collection of grey levels *a*_1_, *a*_2_
*ϵ Z*^+^. As a result of the thresholding operation, a new function is obtained representing the binary image according to the following notation:(25)$$h_{l}\left(x,y\right)=\{\begin{array}{c} a_{1}\;\;dla\;h\left(x,y\right)<t\\ a_{2}\;dla\;h\left(x,y\right)>t \end{array},$$

The resulting binary image has one bit of information: two levels of grey values *a*_1_ and *a*_2_, obtained from thresholding for the value of the *t* threshold and obtained using an arbitrarily selected value. The principle of maximum entropy is one of the tools used for image segmentation by thresholding. Assume that *X* is a discrete random variable, i.e., the range *R* = (*x*_1_, *x*_2_,…) whose set is finite or countable *p_i_* = *P* {*X* = *x_i_*}, *I* = l, 2, …, *n*. The Shannon entropy is defined by Equation (26) [43]:(26)$$H\left(X\right)≅H\left(p_{1},\ldots ,p_{n}\right)≅-\overset{n}{\underset{i=1}{\sum }}p_{i}\cdot \log p_{i},$$
where the random variable *X* represents the uncertainty measure of the stochastic field and *H*(*X*) represents its probability distribution function *p*_1_, *p*_2_,…, *p_n_*.

According to a study [44] on image processing, image *A* can be considered a source of information to build a histogram that determines the desired probability *p_i_*. Therefore, for a given image *A*, that is a set of pixels of the image, and for the resulting probability distribution *p_i_* (26), a threshold *t* is obtained by minimising the described function of the histogram:(27)$$t=\begin{array}{c} \arg\min H\left(X\right)\\ t\;\epsilon \;Z^{+} \end{array},$$

From the newly obtained image, the location of the FOL gauge can be determined, in addition to the specimen’s contour.

### 2.4. VEIP Calibration System

Mechanical calibration is foreseen when transforming the vision camera system into the sample system. The camera’s optical axis is perpendicular to the surface of the image from the camera and runs along the Z-axis to the centre of the sample system (Figure 9). In the first step of the system’s transformation with the camera, geometric conversions consisting of two rotations and one translation are predicted. First, a rotation about the X-axis (angle β) is expected, followed by a translation along the Y-axis and an additional rotation about the Z-axis (angle γ). The corrections of the camera settings in this mathematical model were based on geometric relations illustrated in Figure 9 and the following formula:(28)$$w^{T}=\left[R_{\left(\beta \right)}]\left[T\right][R_{\left(\gamma \right)}\right]W^{T},$$
where the system before rotation is represented by coordinates *W* = (*X*, *Y*, *Z*), and that after the corrections (i.e., the calibration) is represented by *w* = (*x*, *y*, *z*).

The measurement procedure involves selecting the reading’s minimal value from the base length and the initial width obtained during the calibration process. Assume that the sampling length *L* is the distance between a pair of points, *A* and *B*, according to Equation (28), with coordinates {*x*, *y*} that describe the coordinates of the pixels for the digitised image *Z* at the moment of measurement, according to the following equation:(29)$$L=\left|\bar{AB}\right|,$$

Then, during the transformation process (Equation (28)), which generates *n* images, a length *L^+^* is searched in the set of determining distances *D* = *L*_1_, …, *L_n_* for which the following condition is satisfied:(30)$$L^{+}=\min D.$$

A series of tests conducted on static and dynamic images is proposed to determine the magnitudes of the errors resulting from the camera setup in various sample configurations. This allows the determination of the algorithm’s accuracy for the CIPI analysis, the influence of vibrations on the error magnitudes, the impact of the CCD camera setup (calibration process), and the FOL gauge quality. For this purpose, a sample with mounted fibre-optic cables was placed on the lower part of the grip of the machine. The CCD camera was mounted on a specially constructed handle, allowing the camera’s optical axis to be adjusted with respect to the sample surface (Figure 9).

Since the base length and distance between the two FOL gauges are measured in real time, it is possible to track the changes in the parameters of the camera settings and the operation of the calibrating device in real time. Four types of measurement statuses are distinguished in Figure 10: the sample at the stop position (static images to examine the accuracy of the FOL gauge), the static selection during the operation of the machine (dynamic images for use in inspecting the influence of vibrations), displacement of a sample without deformation (dynamic images for use in examining the influence of the CCD camera setup), and change in the operation of the machine (dynamic images illustrating different crosshead speeds and changes in the direction of displacement). The maximum error obtained during the image recording was 0.2 mm, which corresponded to a relative error of 0.2% of the final deformations for a base length of 100 mm.

### 2.5. DIC Technique

The next part of the research into tensile testing focused on measuring deformations using digital image correlation (DIC). Digital image processing through correlation is a vision system for measuring deformations on the surfaces of tested objects. The essence of this method is the maximisation of the normalised correlation factor, according to the following equation:
(31)$$C_{\mathrm{xy}}=\overset{M}{\underset{i=-M}{\sum }}\sum_{j=-M}^{M}\left[\frac{f\left(x_{i},y_{j}\right)g\left(x_{i}^{\prime },y_{j}^{\prime }\right)}{\;\bar{f}\bar{g}}\right],$$
where f (x_i_,y_i_) and g ($x_{i}^{\prime }$, $y_{j}^{\prime }$) are greyscale images for points (x_i_,y_i_) and ($x_{i}^{\prime }$, $y_{j}^{\prime }$) representing the image before and after deformation and functions $\bar{f}$ and $\bar{g}$ define the average greyscale according to the following notation:(32)$$\bar{f}=\sqrt{\overset{M}{\underset{i=-M}{\sum }}\sum_{j=-M}^{M}{\left[f\left(x_{i},y_{j}\right)\right]}^{2}},$$
(33)$$\bar{g}=\sqrt{\overset{M}{\underset{i=-M}{\sum }}\sum_{j=-M}^{M}{\left[g\left(x_{i}^{\prime },y_{j}^{\prime }\right)\right]}^{2}},$$

Calculations are performed for greyscale images with defined subareas with designated central points. The degree of displacement is assessed by comparing two consecutive images before and after deformation. Acquiring high-resolution images is vital as it permits differentiation of the analytical features of an image (e.g., the stochastic grid). The resolution of patterns on the stochastic grid must be consistent with the resolution of the camera capturing the image. The magnitude of displacement in the form of vectors u and v for each point of the stochastic grid is then determined as the difference in the coordinates of pixels for central points of defined subareas before and after deformation. The analysis of subareas, and their monitoring to measure displacement in the correlation method, are used to obtain high-accuracy measurements. To implement the described DIC procedure, a specially designed programme in the Matlab environment was used to generate a mesh grid and a final deformation calculation for a given stochastic pattern [45].

## 3. Results and Discussion

The outcome of uniaxial tensile testing was recorded for all three methods of measuring displacement as a function of the force applied during testing (Figure 11). The accompanying graph illustrates the course of the tests. Two displacement measurements were performed using FOL gauges compatible with the tested material (DC04). Two more tests were conducted to verify the previous results using highly accurate analogue gauges. These readings were taken only for seven selected locations on the samples. The results of the analogue method confirmed the accuracy of the proposed VEIP technique using FOL gauges. Furthermore, the information obtained from the digital gauges demonstrated the influence of the rigidity and reliability of the gripping system. The sample gripping system caused a large discrepancy in the initial phase (spring-back deformation).

Image processing was carried out, according to the aforementioned CIPI analysis procedure, and its fitting within diffuse necking (Figure 1b) was conducted using the GCF process (Figure 12a,b). According to this procedure, for all saved images, calculations were made to identify the specimen contour and then its breadth, thickness, and length within the necking were calculated (Figure 3). Only for the diffuse necking phase was the fitting process used to characterise the geometry’s parameters (Equation (24)), which led to the determination of the stress–strain curve correction factor, according to Equation (17).

As a result of application of the DIC technique using Matlab [42], a series of images displaying the necking phenomenon was obtained for a specimen with an imprinted grid. The final state of the deformation is presented in Figure 13a. As seen in the image, the imprinted grid applied to the specimen incorporated the traditional method of coordinate grid analysis.

However, in the case of stochastic grids, it was possible to define the size of the subdivisions (specify the size of the grid) freely. The number of subareas depended on the computing ability of the software and the available computer memory. The number of images registered during the tensile test (approximately 300) affected the computational process, as increasing the number of images led to more accurate results. If the displacement of points on the grid in consecutive frames was too large, tracking the position of points was hindered, which was a significant issue with this technique. Therefore, remeshing was used whenever the location of points in subsequent images could not be accurately determined. Measurements were made at points on the neck of the sample, both parallel and perpendicular to the direction of elongation, according to the description in Figure 13b,c. Three strains were compared: the true strains parallel and perpendicular to the elongation of the direction of the specimen elongation and the thickness strain (as a result of plastic incompressibility), as described by Equation (5) and shown in Figure 14a. To compare the effects of the DIC, CIPI analysis, and VEIP technique using FOL gauges, the longitudinal and latitudinal strains were examined (Figure 14b).

A slight difference in latitudinal strain between the DIC and CIPI methods arose from differences in the measuring bases. For the DIC technique, the segment was located in the middle of the neck (Figure 13c), while for the CIPI analysis, it was the total breadth of the neck (Figure 12a).

To verify the thickness calculations from the CIPI analysis, measurements of the specimen’s thickness were taken with a specially designed apparatus with digital sensors located on both sides of the specimen. The thickness results were then compared with the calculated thickness from the CIPI analysis (Figure 15).

According to Equation (24), both thicknesses (i.e., *t*, the instantaneous thickness during the diffuse necking phase, and *t_u_*, the thickness at the onset of necking) needed to be calculated to apply the correction factor (Equation (17)). A graphical representation of the correction factor is shown in Figure 16, based on the parameters of diffuse necking (*t*, *t_u_,* and *l*). As the figure shows, the correction for the tensile test of the DC04 material was close to 1 within the diffuse necking phase (the last 20 frames before the onset of localised necking), and no modification of the stress–strain curve was needed. The stress–strain curves obtained with the DIC technique and CIPI analysis were compiled. The results of the methods were consistent, which allowed the designation of the extended stress–strain curve. Finally, the precisions of the vision system (VEIP technique), the CIPI analysis, and the GCF process (up to the final phase of diffuse necking) were evaluated (Figure 17).

Extended DIC computations were performed to examine the deformation of the specimen in the zone with holes drilled for the FOL gauge. Figure 18a shows an image of the sample’s surface and FOL gauge hole location. Figure 18b shows a diagram of the design grid of evaluation coordinates. Figure 18c shows the distribution of the strain field in the vicinity of the FOL gauge hole. The true strain distribution in the vicinity of the fibre showed that the weakening of the specimen, due to the FOL gauge hole location, did not affect the plastic deformation changes when the maximum strain was within the elastic range.

## 4. Conclusions

This paper proposed a simple method for measuring specimen deformation during uniaxial tensile testing using FOL gauges and vision technology (the VEIP technique). Elongation values determined in this manner were verified and compared with those obtained with traditional measurement devices, such as analogue extensometers and digital gauges. Deformation measurement using FOL gauges and the VEIP technique and an advanced correlation technique (DIC) proved to be a quick and straightforward way to determine the stress–strain curve for uniform deformation. The strain field distribution was also verified in the FOL gauge zone, and no significant deformations were observed. CIPI analysis and the GCF process are proposed as a precise and rapid method to determine the true stress–strain curve for diffuse necking.

To obtain accurate results for deformation during necking, the authors proposed verification of the stress correction factor by describing the changes that occur in necking geometry through Gaussian functions and calculating the specimen thickness, by applying the assumption that the strain components in the latitudinal and thickness directions were equal. The correction factor calculations showed that no stress–strain curve modification was needed for the diffuse necking phase. The proposed solution is an alternative to the DIC technique that involves less advanced computations (i.e., analysis of individual points and contour identification only).

## Figures and Tables

**Figure 1 materials-16-00558-f001:**
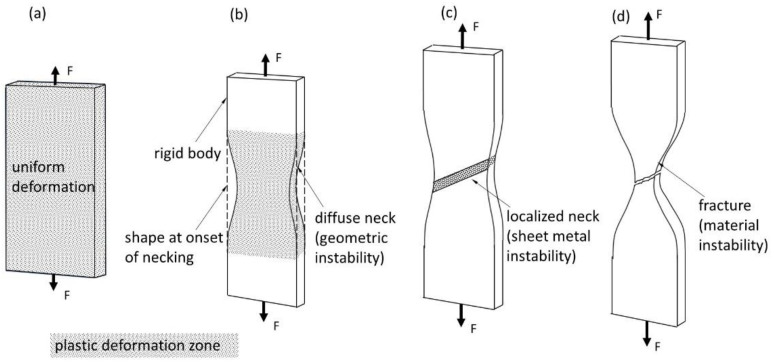
Tensile test phases in a flat specimen: (**a**) uniform deformation, (**b**) geometric instability, (**c**) localised necking, (**d**) cracking.

**Figure 2 materials-16-00558-f002:**
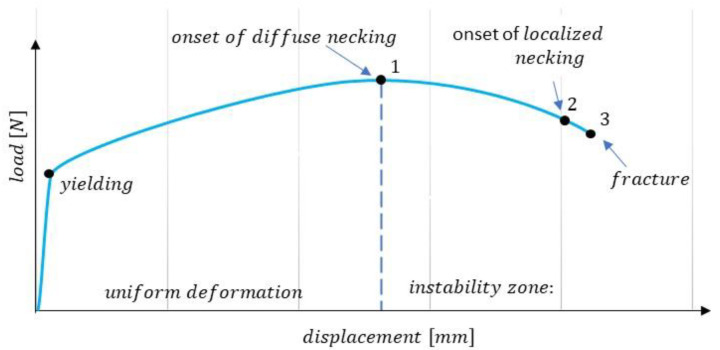
Load–displacement curve illustrating phases of the tension test.

**Figure 3 materials-16-00558-f003:**
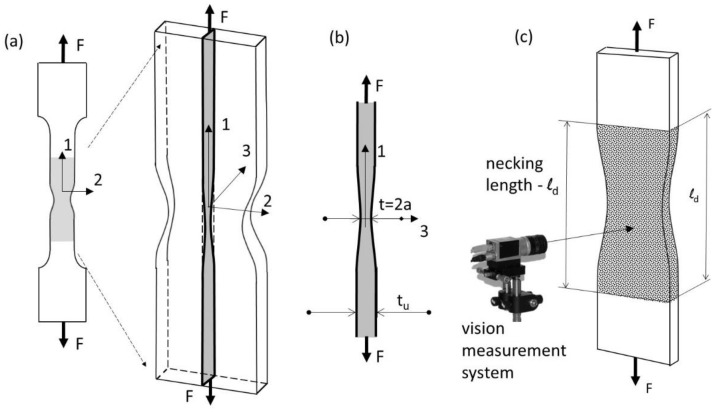
Uniaxial tension test. (**a**) plane strain model for necking, (**b**) the cross-section represent thickness of the sample, (**c**) schematic diagram of the vision measurement.

**Figure 4 materials-16-00558-f004:**
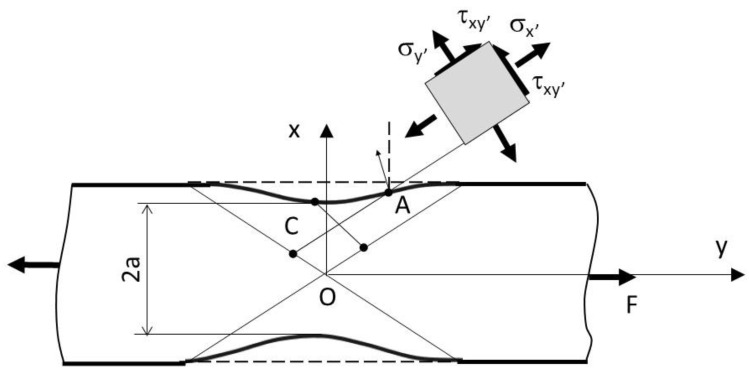
Geometric interpretation of the strain identification, following [29].

**Figure 5 materials-16-00558-f005:**
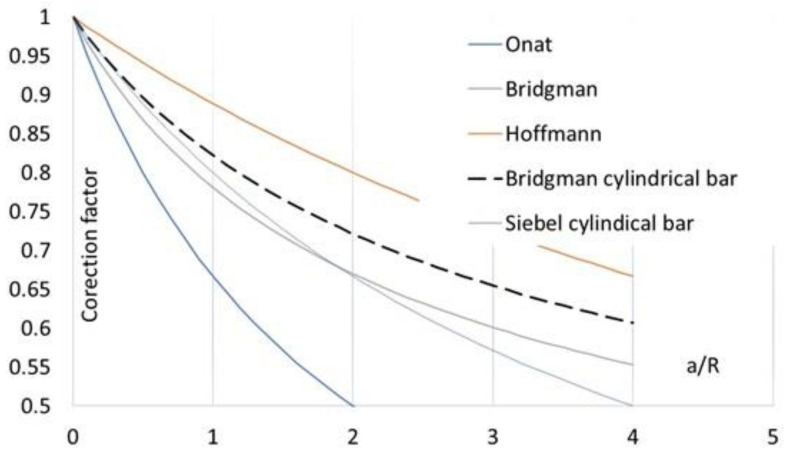
Comparison of stress correction parameters.

**Figure 6 materials-16-00558-f006:**
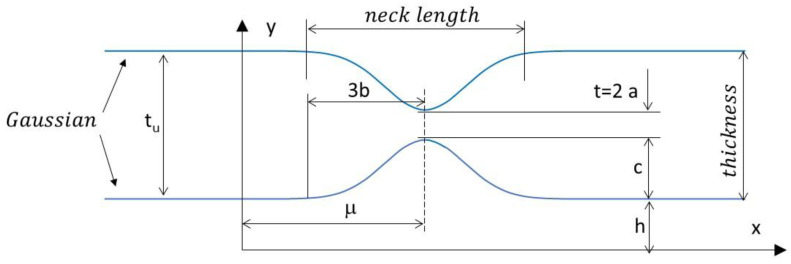
Schematic illustration of Gaussian contour fitting within the diffuse necking zone.

**Figure 7 materials-16-00558-f007:**
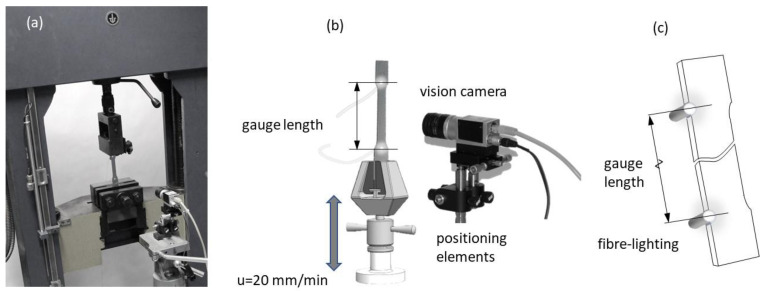
The proposed vision extensometer technique illustrations: (**a**) tension test measurement, (**b**) schematic illustration of the vision extensometer, (**c**) gauge length illustration.

**Figure 8 materials-16-00558-f008:**
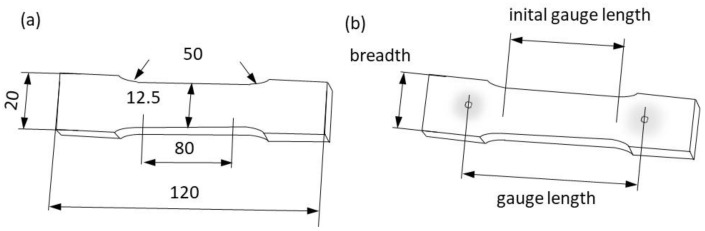
Specimen illustrations: (**a**) sample dimensions, (**b**) schematic illustration of gauging.

**Figure 9 materials-16-00558-f009:**
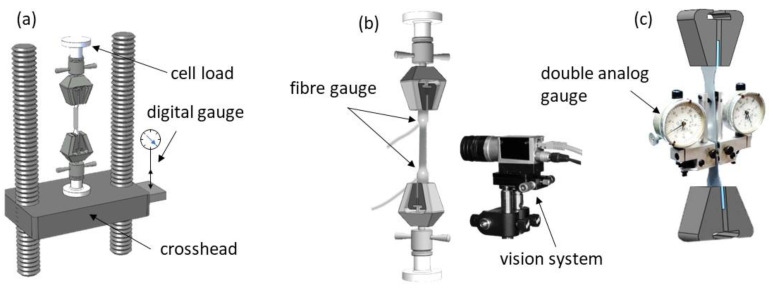
Schematic view of three techniques for acquisition of load and deformation data during uniaxial tensile testing: (**a**) load sensor and digital gauge located on the crosshead, (**b**) vision extensometer, (**c**) analogue extensometer.

**Figure 10 materials-16-00558-f010:**
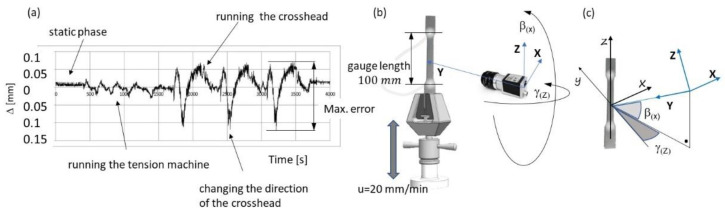
Accuracy measurement, (**a**) measurement results for different conditions, (**b**) measuring system with calibration, and (**c**) system transformation characteristics.

**Figure 11 materials-16-00558-f011:**
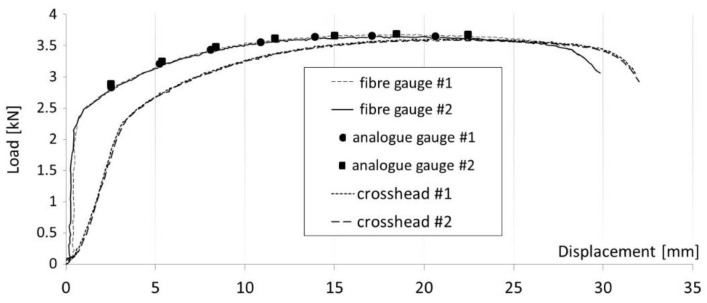
Load–displacement results for three different measurement techniques.

**Figure 12 materials-16-00558-f012:**
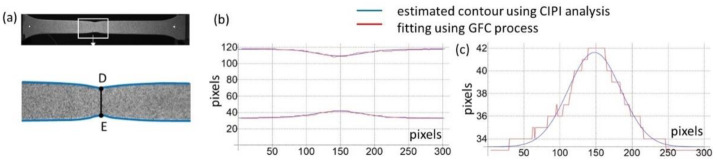
Identification of the specimen’s contour outline: (**a**) view of the specimen identification, (**b**) contour recognition using image processing, (**c**) detailed fitting using Gaussian curve.

**Figure 13 materials-16-00558-f013:**
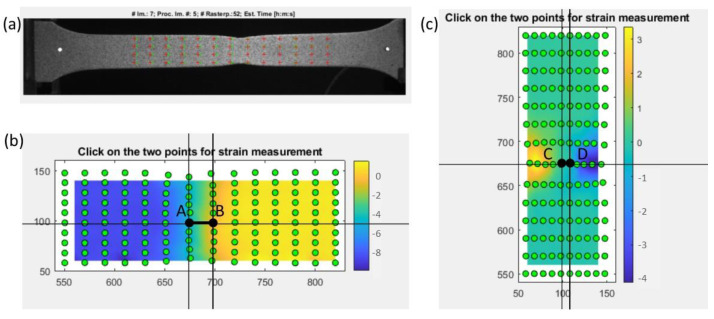
Characteristics of the DIC measurement: (**a**) final stage of tension, (**b**) true strain measurement in the longitudinal direction (distance AB), (**c**) true strain measurement in the latitudinal direction (distance CD).

**Figure 14 materials-16-00558-f014:**
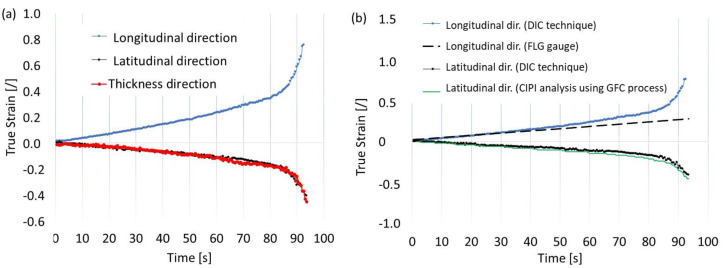
Comparison of results for the strain components: (**a**) for the DIC technique, (**b**) for the image processing using fibre-lighting gauge and contour measurement.

**Figure 15 materials-16-00558-f015:**
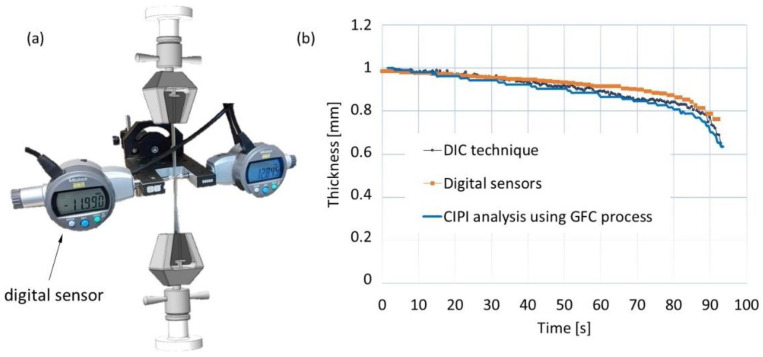
Thickness measurement: (**a**) designed thickness measurement gauge, (**b**) comparison of the thickness measurement (digital gauge) and thickness calculation using DIC technique and CIPI analysis.

**Figure 16 materials-16-00558-f016:**
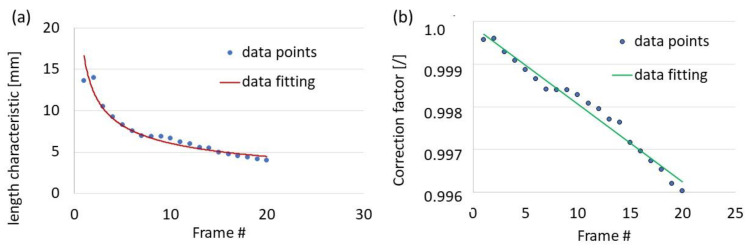
Correction of the true stress–strain curve calculation: (**a**) distribution of the necking length parameter characteristic for the last 20 frames of recorded tension test, (**b**) distribution of the correction factor.

**Figure 17 materials-16-00558-f017:**
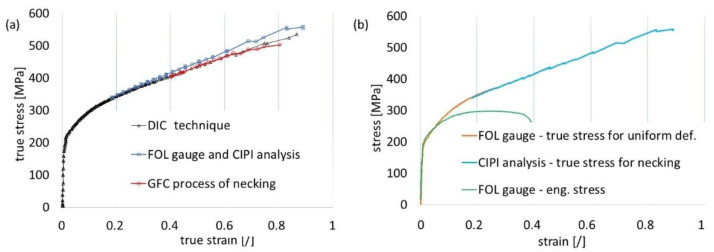
Comparisons of results for the true stress–strain curve: (**a**) DIC and image processing, (**b**) image processing using fibre-lighting and contour calculation.

**Figure 18 materials-16-00558-f018:**
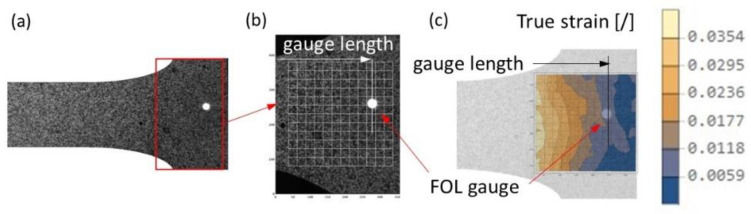
Strain field measurement using DIC: (**a**) specimen view, (**b**) design grid, (**c**) true strain distribution for the stretched area.

**Table 1 materials-16-00558-t001:** Chemical composition of the material—DC04.

Material	C [%]	Mn [%]	P [%]	S [%]	F [%]
DC04	0.055	0.25	0.01	0.008	99.5

## Data Availability

Data sharing not applicable.

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
