# Peer review of "A New Approach for Evaluation True Stress–Strain Curve from Tensile Specimens for DC04 Steel with Vision Measurement in the Post-Necking Phases"

_materials, 2023, doi:10.3390/ma16020558_

Round 1
Reviewer 1 Report
Authors presented a method to characterize the flow curve beyond the diffuse necking of sheet metals using a tensile test.
The manuscript repeats many simple and well-known facts from textbooks, like details of strain and stress descriptions, tensile test etc. These should be avoided. On the other hand, the most important part, which is the contribution of authors, is not explained very detailed. It requires some more pictures and explanations to put the provided pictures and equations together to understand the novel development.
Furthermore, the literature provided is not complete or at least does not include the most important recent publication in the field of "characterization of flow curve beyond diffuse necking".
The validity of the approach is tested using a single material, which has a long elongation in the diffuse necking phase. At least one other material with a short diffuse necking phase should also be considered to evaluate the proposed approach.
Author Response
Thank you very much for all comments and suggestions. In reference to them, a new section: “2.1. Vision extensometer techniques” was added. In this section a proposed techniques has been describes in detail with a new illustration (Fig. 7).
The stress correction was successfully incorporated into the round bar tension test as the authors pointed out. However, for flat samples the correction factor is continue to be proposed, and new formulas are still being investigated. For the flat samples the neck contour is less deformed, smooth, asymmetrical, and challenging to identify, therefore many investigators have faced the problem of accurately characterising the necking geometry. Some methods are complicated and not applicable in practice according to the literature.
As for the validity of the approach, which is tested using a single material, the presented paper contains a very extensive study of the theoretical part and therefore less emphasis was placed on extended experimental research (as noted by the reviewer). The authors proposal techniques foresees the extension of the experimental investigation in the next paper.
A literature survey has been updated according to reviewer suggestions, concerning either tensile test analysis methods and materials characterization models.
Reviewer 2 Report
The paper that I have reviewed it today sounds to be consistent. It is little bit abstract in the way that formulas are explained and presented in the text!
It would be interesting (this is just a recommendation) and maybe much attracting to include some aspects related to numerical simulation / FEA analyses performed in the text instead of just using lot of mathematics formulas and so on. But I want to repeat - this is just a recommendation or a suggestion!
I really appreciated much more the practical part of the research how it was presented and emphasized in the text - and how the proposed solution it aims to be an alternative to DIC technique with less computations in the end (I was conviced about it more in the second part of the article about it)!
Regarding the formal things / editing of the paper I can say that the paper is quite well written in English language, just some headlines about few figures (especially Fig. 3 and 17) should be moved under the presented images, not below.
Also maybe the title could be reconsidered little bit! In the way it is formulated now: "Tensile test evaluation using a vision extender and image processing" it is not as much consistent as the paper it is!
Tensile test evaluation of what? Maybe can be specified / added in the title somehow that the evaluation was made in the case of DC04 steel, that as it is stated in the paper, it is commonly used in the automotive 370 industry
Also in the title somehow has to be added that the proposed method of evaluation of using a vision extender and image processing it is a new one!
The evaluation it is not so clear by just reading the title about what is related to!
Also in terms of References these are quite good as they are provided in the text, but many of these references are quite old (more than 30 years ago many of them), just one reference is dated in 2020 and one in 2022...
Somehow, as a general impression it would be great if at least 10 references could be extra added / to other similar scientific papers that were published by MDPI journals (Materials MDPI journal in particular) in the period 2020-2022.
Based on this observations that were stated above, I consider that the paper can be accepted in the MDPI Materials journal after minor revisions that have to be done as they were recommended above.
Author Response
Regarding the formal things / editing of the paper I can say that the paper is quite well written in English language, just some headlines about few figures (especially Fig. 3 and 17) should be moved under the presented images, not below.
The headlines from Figures 3 and 17 have been moved under the presented images.
Also maybe the title could be reconsidered little bit! In the way it is formulated now: “Tensile test evaluation using a vision extender and image processing” it is not as much consistent as the paper it is!...
With regard to the suggestions made, the title has been modified as follows:
“A new approach for evaluation true stress–strain curve from tensile specimens for DC04 steel with Optical Strain Measurement in the post-necking phases”
Also in terms of References these are quite good as they are provided in the text, but many of these references are quite old (more than 30 years ago many of them), just one reference is dated in 2020 and one in 2022...
In the presented descriptions of measurement techniques, the authors refer to the early researches of these solutions, whose participation in the development of methods is unquestionable.
Somehow, as a general impression it would be great if at least 10 references could be extra added / to other similar scientific papers that were published by MDPI journals (Materials MDPI journal in particular) in the period 2020-2022.
A literature survey has been updated according to reviewer suggestions, concerning either tensile test analysis methods and materials characterization models.
Reviewer 3 Report
· Abstract is not on point, also add new techniques and modifications which you Implied in the manuscript.
· Introduction looks short and crisp yet need some more number of references should be added.
· Focus more on application-oriented papers and also use different tensile test evaluation comparative results.
· kindly provide more information on vision extender and image processing.
· The explanation of the materials and methods and the sequence of some information presented should be improved to make the article clear to be understood by the readers.
· Explain the reason for improved properties in the results and discussion parts. Graph needs to be more detailed.
· All the properties and tables should be incorporated in the manuscript.
· Many are very past references. Please use references that are 2018 and newer.
· Manuscript has some grammatical errors and punctuation errors kindly correct the Whole manuscript.
Author Response
Thank you very much for all comments and suggestions. In reference to them, an article has been partially rewritten and improved.
- Introduction looks short and crisp yet need some more number of references should be added.
A literature survey has been updated according to reviewer suggestions, concerning either tensile test analysis methods and materials characterization models.
- Focus more on application-oriented papers and also use different tensile test evaluation comparative results.
The presented paper contains a very extensive study of the theoretical part and therefore less emphasis was placed on extended experimental research (as noted by the reviewer). The authors proposal techniques foresees the extension of the experimental investigation in the next paper.
- kindly provide more information on vision extender and image processing.
- The explanation of the materials and methods and the sequence of some information presented should be improved to make the article clear to be understood by the readers.
A new section: “2.1. Vision extensometer techniques” was added. In this section a proposed techniques has been describes in detail with a new illustration (Fig. 7).
- Explain the reason for improved properties in the results and discussion parts. Graph needs to be more detailed.
The new graphic has been added and remained partially modified.
- All the properties and tables should be incorporated in the manuscript.
In the presented solutions, the main goal is to indicate an analytical solution as well as experimental procedure. Therefore, the formula proposed by the reviewer for the summary of results in the form of table was not used.
- Many are very past references. Please use references that are 2018 and newer.
Newer positions of the literature survey has been added.
- Manuscript has some grammatical errors and punctuation errors kindly correct the Whole manuscript.
The whole manuscript has been reviewed in detail to correct grammatical errors and punctuation errors as well. Summary of this task is presented in PDF file.
Reviewer 4 Report
Reviewer comments
This paper stated on the “Tensile test evaluation using a vision extender and image processing”.
1- The title of paper is too simple and not clear. The tensile test evaluation of what? What kind of image processing?
2- The section 1 (Introduction section), more citation related to the tensile test evaluation of flat samples should be added.
3- The section 1 is too long. It is very difficult for the readers to find the main highlight and purpose from this paper.
4- Make another section for subsections: 1.1. Extensometer measurement techniques, 1.2. Theory of the tension test
5- The quality of images in the Figures is too low. Improve the quality of images. Make them clearer.
Author Response
Thank you very much for all comments and suggestions. In reference to them, the article has been rewritten partially and improved. A few answers for the reviewer questions/suggestions are as follwos:
- The title of paper is too simple and not clear. The tensile test evaluation of what? What kind of image processing?
The title has been modified as follows:
“A new approach for evaluation true stress–strain curve from tensile specimens for DC04 steel with Vision Measurement in the post-necking phases”
- The section 1 (Introduction section), more citation related to the tensile test evaluation of flat samples should be added.
A tensile test evaluation citations have been added.
- The section 1 is too long. It is very difficult for the readers to find the main highlight and purpose from this paper.
The section was modified and the aim of the research underlined more precisely.
- Make another section for subsections: 1.1. Extensometer measurement techniques, 1.2. Theory of the tension test
A new section: “2.1. Vision extensometer techniques” was added. In this section a proposed techniques has been described in detail with a new illustration (Fig. 7). Therefore, the numbering of images from Fig.7 has been updated.
- The quality of images in the Figures is too low. Improve the quality of images. Make them clearer.
The images were modified as follows:
Fig. 3 the vision system was replaced to make clear identification of the illustration,
Fig. 8. A last part of the figure (c) was removed and described in detail in a new figure 7.
Fig. 10b. was modified to and the coordinate system was assigned to the camera for better link with Figure 10c
Round 2
Reviewer 1 Report
There are very minor issues to be corrected.
Typing error "Hollemon" should be "Hollomon" for example.
Reviewer 3 Report
The authors revised as per the direction, and it can be accepted in the present form
Reviewer 4 Report
The quality of paper has been improved.
It can be accepted for the publication.